# Spatial distribution of mixed milk feeding and its determinants among mothers of infants aged under 6 months in Ethiopia: Spatial and geographical weighted regression analysis

Mekuriaw Nibret Aweke[1]*, Muluken Chanie Agimas[2], Moges Tadesse Abebe[3], Tigabu Kidie Tesfie[2], Meron Asmamaw Alemayehu[2], Werkneh Melkie Tilahun[4], Gebrie Getu Alemu[2], Worku Necho Asferie[5]

1 Department of Human Nutrition, Institute of Public Health, College of Medicine and Health Sciences, University of Gondar, Gondar, Ethiopia, 2 Department of Epidemiology and Biostatistics, Institute of Public Health, College of Medicine and Health Sciences, University of Gondar, Gondar, Ethiopia, 3 Department of Nursing, College of Health Science, Debark University, Debark, Ethiopia, 4 Department of Public Health, College of Medicine and Health Sciences, Debre Markos University, Debre Markos, Ethiopia, 5 Departments of Pediatrics and Neonatal Nursing, College Health Science, DTU, Debre Tabor, Ethiopia

* mekunib@gmail.com

## Abstract

### Background

Mixed milk feeding is defined as providing formula and/or animal milk along with breast milk to infants under six months old which is prevalent in many countries. However, this practice is generally not recommended as it can reduce the intake of breast milk, depriving the infant of its optimal nutritional and immunological benefits. Unlike formula, breast milk contains complex bioactive constituents that promote intestinal and pancreatic growth and develop mucosal defenses. The aim of this study was to analyze the spatial distribution and predictors of MMF practices in Ethiopia.

### Methods

This study utilized data from the 2019 Mini-Ethiopian Demographic and Health Survey (MiniEDHS), a nationally representative cross-sectional survey conducted from March to June 2019. The total weighted sample size derived from the data examined in this study amounted to 524 infants. The data analysis used Global Moran's I for spatial autocorrelation and the Getis-Ord Gi* statistic for local cluster analysis to assess the spatial distribution of mixed milk feeding prevalence across Ethiopia's administrative regions and cities. Empirical Bayesian Kriging was used for spatial interpolation to estimate mixed milk feeding prevalence in unsampled areas. The analysis utilized a maximum spatial cluster size threshold of 50% of the population to detect clusters of varying sizes. Ordinary least squares regression analysis identified significant spatial predictors. In geographically

**Data availability statement:** Third party data was obtained for this study from the DHS Program (Ethiopian Mini Demographic and Health Survey (miniDHS) 2019). Data may be requested from the DHS Program after creating an account and submitting a concept note. More access information can be found on the DHS Program website (https://dhsprogram. com/data/Access-Instructions.cfm). The authors confirm that interested researchers would be able to access these data in the same manner as themselves. The authors also confirm that they had no special access privileges that others would not have.

**Funding:** The author(s) received no specific funding for this work.

**Competing interests:** The authors have declared that no competing interests exist.

weighted regression analysis, the effect of predictor variables on the spatial variation of mixed milk feeding was detected using local coefficients.

## Results

The overall weighted prevalence of Mixed Milk Feeding (MMF) in Ethiopia was 10.12% (95% CI: 7.8, 13.01). This prevalence shows significant regional variations across the country emphasizing regional disparities in prevalence and distribution. The Global Moran's I statistic was 0.14, with a Z-score of 3.18 and a p-value of <0.001, indicating a significant spatial clustering of MMF prevalence. Hotspots of mixed milk feeding were identified in Somali, Dire Dawa, and Afar, while cold spots were observed in Amhara, Tigray, Benishangul Gumuz, SNNPR, and parts of Oromia. Household wealth (middle wealth index) and lack of baby postnatal checkups emerged as key influencers of mixed milk feeding practices.

## Conclusion

The study found significant regional variations in mixed milk feeding practices in Ethiopia. Households with middle wealth index and baby without postnatal check were significant spatial predictors of mixed milk feeding. To reduce mixed milk feeding prevalence, targeted interventions should engage community leaders, enhance breastfeeding education in maternal health services, and integrate counseling into routine healthcare to support informed maternal choices and improve child health outcomes nationwide.

## Background

The World Health Organization (WHO) advocates for exclusive breastfeeding (EBF) for infants under six months of age [1]. This practice involves providing only breast milk to infants, excluding all other foods and beverages, except for oral rehydration solutions and necessary supplements [1,2]. Breast milk provides all the energy and nutrients needed for the first six months of life, continues to supply up to half or more of a child's nutritional needs during the second half of the first year, and up to one third during the second year [3].

Furthermore, breastfeeding offers well-documented benefits for nutrition, health, and development, and has positive emotional and psychological effects, including behavioral implications [4].

Encouraging breastfeeding could avert 823,000 child deaths annually, accounting for 13.8% of all child fatalities [5]. Moreover, breastfeeding has the potential to prevent nearly half of all diarrhea cases and one-third of respiratory infections in children [5,6].

Despite breast milk being the ideal food for infants, many do not receive it for the recommended duration due to the use of alternative milk beverages like infant formula and animal milk products [7]. More than 85% of mothers worldwide do not follow the WHO recommendation, and only 35% of infants younger than four months are exclusively breastfed [8]. Evidence shows that while most mothers start with EBF, the rate declines significantly after two or more months [9]. Over a million children's lives could be saved, and their quality of life significantly enhanced by EBF [10–12].

Mixed milk feeding (MMF), defined as the provision of formula and/or animal milk in addition to breast milk to infants under six months of age, is a common practice in many countries [13]. However, this practice is generally not recommended as it can reduce the

intake of breast milk, depriving the infant of its optimal nutritional and immunological benefits [14]. Unlike formula, breast milk contains complex bioactive constituents that promote intestinal and pancreatic growth and develop mucosal defenses [15,16]. Formula-fed babies are more susceptible to various health issues, including diabetes mellitus, otitis media, allergies, gastroenteritis, pneumonia, diarrhea, reduced cognitive development, increased risk of obesity, and sudden infant death syndrome [17,18].

MMF in infants under six months can lead to early termination of breastfeeding, reduced breast milk production, and changes in gut microflora [13,19]. Studies have shown that a reduced proportion of breast milk in combination with other feeds can notably increase the risk of infections, such as gastrointestinal and respiratory infections, as well as a higher risk of developing allergies, asthma, and other chronic conditions later in life [20,21]. Studies have shown that breastfeeding improves the emotional and psychological connection between mother and infant [22,23]. Therefore, MMF infants lose this critical bonding experience [24].

In Ethiopia, infant mortality remains a significant burden, with suboptimal breastfeeding practices contributing to a large number of infant deaths annually [25]. Inadequate breastfeeding, including MMF, leads to unnecessary child deaths, illnesses like diarrhea and pneumonia, and significant household and healthcare costs [26]. The mother's educational status, attendance at postnatal care, marital status, place of delivery, and media exposure, household income status, child's age, residence, occupation, time of initiation of breast feeding were all found to be predictors of exclusive breastfeeding practice in the previous studies [27–29].

By 2030, the Sustainable Development Goals (SDGs) aim to decrease newborn and under-five mortality rates to 12 and 25 per 1,000 live births, respectively [30]. To achieve this, the Global Breastfeeding Collective, led by UNICEF and WHO, mobilizes support across political, legal, financial, and public spheres for breastfeeding. Additionally, WHO and UNICEF provide specialized training for healthcare professionals to offer skilled assistance to breastfeeding mothers, address challenges, and monitor children's growth, facilitating early detection of undernutrition or overweight/obesity risks [31].

Currently, Ethiopia is implementing interventions and strategies to promote optimal breastfeeding, including community education programs, healthcare provider training, and policy implementation to create supportive environments for breastfeeding mothers [32]. Despite the substantial efforts to promote optimal breastfeeding in Ethiopia, the prevalence of optimal breastfeeding remains low, posing a significant challenge to public health initiatives [33].

Due to the increasing practice of MMF in all countries, including middle- and low-income nations, WHO and UNICEF have developed a new indicator which has been included to capture the practice of feeding formula and/or animal milk in addition to breast milk among infants less than six months of age [13]. This indicator is used to examine the status of MMF in different countries, including Ethiopia to improving breastfeeding practices, reducing child morbidity, and achieving Sustainable Development Goals related to child health. To this day, various research has been carried out in Ethiopia to evaluate the exclusive breast feeding among children, including the bottle feeding [8,34,35].

Nonetheless, there has been no research conducted to determine the geographical distribution of MMF and its products across the regions of Ethiopia. Analyzing the spatial distribution of MMF and its prevalence across Ethiopian regions is essential for implementing targeted local interventions aimed at reducing undernutrition, child morbidity, and mortality associated with suboptimal breastfeeding practices. Therefore, the aim of this study was to assess the spatial distribution of MMF and the associated factors among children under six months of age in Ethiopia. This research will contribute to the existing body of knowledge and inform policy-making to improve infant nutrition and health outcomes in Ethiopia and similar settings.

## Methods and materials

### Data source

The study conducted a community-based cross-sectional analysis using data from the 2019 Mini-Ethiopian Demographic and Health Survey (MiniEDHS). The survey was conducted from March 21, 2019, to June 28, 2019, based on a nationally representative sample that provided estimates at the national and regional levels and for urban and rural areas. Samples of enumeration areas were stratified and selected in two stages based on the 2019 DHS data collection methods [36]. In the first stage, 305 enumeration areas (93 urban and 212 rural areas) were selected. In the second stage, households were selected. Data were requested online from the international DHS and accessed from the official website of DHS (www.dhsprogram.com). The detailed methodology of the survey is described in the Mini-EDHS report. Total weighted samples of 524 live infants less than 6 months of age who were the youngest and lived with their mother were included in the analysis. Missing data were managed based on the DHS guideline. Ethiopia, located in the Horn of Africa, is known for its diverse topography and positioned in the horn of Africa, spanning from 30 to 140N and 330 to 48°E.

### Sampling procedures and populations

All living infants aged under 6 months constituted the source population, with the study population comprising selected infants living with their mothers. For this study, the research utilized Kids Record (KR) files, which encompass data concerning both women and children. The focus was on extracting significant variables associated with MMF practices from the dataset. The 2019 Mini-EDHS initially included 618 infants under 6 months of age. For our study on MMF practices, we excluded infants who were not breastfeeding. This resulted in a final weighted sample of 524 infants who met the study criteria. Among the total of selected clusters with zero coordinates were excluded from the analysis. In each of the nation's nine administrative regions and two administrative cities, a hierarchical structure is established comprising zones, districts, towns, and kebeles, with kebeles representing the smallest administrative units.

## Outcome variable

### Mixed milk feeding

The outcome variable MMF identifies infants aged 0–5 months who received both breast milk and formula or animal milk within the previous day, defining mixed milk feeding. Infants exclusively fed breast milk were categorized as non-MMF. Thus, infants were categorized as 1 (MMF) or 0 (non-MMF). Finally, a weighted proportion of MMF per cluster was utilized for spatial analysis.

### Independent variables

From the standard 2019 EDHS dataset, mothers' age, age of the infant, number of ANC and PNC visits, household wealth index (poor, middle, rich), mothers' marital status, occupational status, place of residence, place of delivery, mothers' educational level, history of caesarean section, and breastfeeding counselling were candidate predictor variables for the spatial regression model. Whereas, administrative region of the country was taken as community (cluster) level independent variables.

### Data collection and tools

The Mini-Ethiopian Demographic and Health Survey (MiniEDHS) data were collected through face-to-face interviews using questionnaires at the individual and household levels.

During the data collection period, mothers with infants aged 0-5 months were asked to give important socio-demographic, socio-economic status, obstetric, and child-related characteristics that were associated with MMF practices in Ethiopia.

## Data management and analysis

The proportions of the outcome variable and potential predictor variables were tabulated in STATA/MP version 17.0 software (StataCorp LLC, College Station, TX, USA) and exported to excel before being used for further analysis. Before performing analysis, the data was weighted (using sample weight) for MMF practices and candidate explanatory variables. The spatial analysis was carried out with ArcGIS 10.7 and SaTScan V.9.6 software for the local cluster analysis. We remove missing values from our analysis by using the STATA drop command in combination with a logical or conditional statement.

## Spatial analysis

Spatial autocorrelation (Global Moran's I) was used to assess spatial heterogeneity for MMF among infants aged 0–5 months. Global Moran's I is a spatial statistic that measures spatial autocorrelation by taking the entire data set and generating a single output value ranging from -1 to +1. Moran's I value close to -1 indicates that MMF is dispersed, whereas Moran's I close to +1 indicates that MMF is clustered, and Moran's I close to 0 indicates that MMF is randomly distributed. A statistically significant Moran's I value ($p < 0.05$) had a chance to reject the null hypothesis which indicates the presence of spatial autocorrelation. Hot spot analysis (Getis-Ord Gi* statistic) z-scores and significant p-values gave the features with either hot spot or cold spot values for the clusters spatially. A 'hotspot' denotes areas with a high prevalence of clustered MMF on the map, whereas a 'cold spot' indicates areas with a low prevalence of MMF clustering together spatially.

## Spatial interpolation

The spatial interpolation technique used to predict MMF among infants aged 0–5 months for unsampled areas in the country. For the prediction of unsampled Enumeration Areas, we used geo-statistical Empirical Bayesian Kriging spatial interpolation techniques using ArcGIS 10.7 (ESRI Inc., Redlands, CA, USA, version 10.7) software. Empirical Bayesian Kriging relaxes the assumption of the Gaussian distribution of the observed semi-variogram in the input data which rarely holds in practice. Empirical Bayesian Kriging interpolation works by generating a new simulated semi-variogram at each location from the estimated semi-variogram from the input data. The weight of the new simulated semi-variogram was calculated by Bayes' rule [37].

## Spatial scan statistics

We employed spatial scan statistics to identify local clusters of statistically significant high and low rates of MMF practice using SaTScan 10.1. We used the Bernoulli model to identify significant clusters of mixed milk feeding (MMF vs. non-MMF) in infants aged 0–5 months, with a scanning window across the study area [38]. The case (presence of the outcome variable), control (absence of the outcome variable), and coordinate file (latitude and longitude) were imported into the software to identify the locations of significant clusters.. A cluster is statistically significant when its log-likelihood ratio (LLR) is greater than the standard Monte Carlo critical value at a value of p less than 0.05 [39]. The maximum likelihood ratio test statistic showed the most primary cluster relative to the global distribution of maximum values [40].

The primary and next further significant clusters were identified, the LLR was assigned, and the value of p was obtained through Monte Carlo hypothesis testing with 999 Monte Carlo replicates [41]. The default maximum spatial cluster size of less than 50% of the population used as an upper limit, allowing both small and large clusters to be detected, and ignored clusters that contained more than the maximum limit with the circular shape of the window.

## Spatial regression

**Ordinary least squares (OLS).** We applied spatial regression modeling to explore predictors of spatial variation in MMF across the study area. First, we used the Ordinary Least Squares (OLS) model, a global approach that estimates a single coefficient for each explanatory variable over the entire study area [42]. Spatial regression modeling was used to identify predictors of the spatial variation of MMF in the study area. The ordinary least squares regression (OLS) model is a global model that estimates only one single coefficient per explanatory variable over the entire study area. The results of ordinary least squares (OLS) regression are only reliable if all six assumptions are fulfilled. The coefficients of explanatory variables in a specified, properly constructed OLS model should be statistically significant and have a positive or negative sign. The model should be non-stationary, include key explanatory variables and be free from multicollinearity [43]. In addition, residuals should be normally distributed, should not reveal spatial patterns and be free from spatial autocorrelation. A data mining tool was used to identify a model that fulfils the assumption of the OLS regression. Additionally, explanatory regression identified models that fulfilled the assumptions of the OLS methods and models with high adjusted R2 values. The final model was validated by internal cross-validation. Multicollinearity (VIF < 10) was also tested to rule out redundancy among independent variables.

**Geographically weighted regression analysis.** A strong predictor variable in one cluster may not be a strong predictor in another cluster. Such a type of cluster variation (non-stationarity) can be detected using geographically weighted regression (GWR) [44]. OLS uses a single linear regression equation for all of the data in the study area, whereas GWR creates an equation for each cluster. Therefore, the coefficients of GWR take different values for each cluster. The optimal bandwidth for GWR was selected using cross-validation, ensuring a balance between model complexity and accuracy. The model was validated by checking for spatial autocorrelation in the residuals using Moran's I statistic, confirming that the residuals were randomly distributed and free from spatial patterns. The GWR map of the coefficients of each predictor variable guides targeted interventions [45,46]. The GWR model is represented as:

$$yi = B_0 \left(\text{uivi}\right) + \sum_{k=1}^{p} \text{Bk}\left(\text{ui,vi}\right)\text{xik} + \text{Ei}$$

where yi is the observation of response; (uivi) is the latitude and longitude; βk (ui, vi) (k = 0, 1,… p) is the p unknown function of the geographical location (uivi); xik is the independent variable at location (uivi), where i is equal to 1,. 2,…; and εii is the error term/residual with zero mean and homogeneous variance σ. Finally, the best-fitted model for the data was determined based on the lowest AICc score and the highest adjusted R-squared.

## Ethical consideration

Permission for data access was obtained from a major Demographic and Health Survey through an online request at (http://www.dhsprogram.com). The data used for this study were publicly available with no personal identifier. Our study was based on secondary data from Ethiopian Demographic and Health Survey and we have secured the permission letter from the main Demographic Health and Survey.

## Result

### Socio-demographic characteristics of participants

A total weighted sample of 524 infants below 6 months of age participated in this study. Over half of the infants, 280 (53.5%), were female, while under two-fifths, 209 (39.9%), were within the 4–5 months age group. Among the mothers, 125 (23.8%) were aged 30–35, while 134 (25.6%) were in the 25–29 age range. The majority of mothers (491, 93.8%) were married, and nearly half (260, 49.6%) had no formal education. The household head was predominantly male, accounting for 461 (88.0%) of households, whereas only 62 (11.9%) of household heads were female (11.9%). The majority of participants resided in rural areas (403, 77.1%), while 119 (22.9%) lived in urban areas. The three regions with the highest percentage of participants were Oromia (225, 43.0%), followed by SNNPR (Southern Nations, Nationalities, and Peoples' Region) with 95 (18.1%), and Amhara with 93 (17.7%). Based on wealth index tiers, approximately one-fourth (131, 25.0%) of the participants were categorized as poor, 228 (43.5%) as middle, and 166 (31.6%) as rich (Table 1).

### Prevalence of mixed milk feeding in Ethiopia

In this study, the overall prevalence of MMF was 10.12% (95% CI 7.8, 13.01), with regional variations observed across Ethiopia. Dire Dawa reported the highest prevalence at 33.95%, followed by the Afar region at 30.40% (Fig 1). Mixed milk feeding was more common among mothers of infants aged 0-5 months living in urban areas, with a prevalence of 18.06%, compared to 7.76% in rural areas.

### Spatial autocorrelation of mixed milk feeding practice in Ethiopia

The spatial patterns of MMF among mothers of infants aged 0-5 months in Ethiopia were clustered. The global spatial autocorrelation analysis showed that there were significant clustered patterns of MMF in the regions of Ethiopia (Global Moran's I = 0.14, Z-score = 3.18, p-value < 0.001) (Fig 2A). This stated that MMF in Ethiopia with similar patterns were interdependent. The figures below showed the clustered patterns (on the right side) with high rates of MMF across regions in Ethiopia (Fig 2B).

### Hot spot analysis

Hot Spot Analysis was conducted using the Getis-Ord Gi* Spatial Statistics method to identify statistically significant spatial clusters of cold spots and hot spots for MMF across Ethiopia. The results reveal distinct geographic patterns in the prevalence of MMF, highlighting areas with significant clustering of both high and low values.

Statistically significant hotspot areas of MMF, represented by red dots on the map, are predominantly located in the Somali, Afar, and Dire Dawa regions. These regions exhibit a high concentration of maximum MMF values, indicating that they have a significantly higher prevalence of MMF compared to other regions. Conversely, the analysis identified significant cold spot areas of MMF, marked by blue dots, in most parts of Amhara, Tigray, Benishangul Gumuz, SNNPR and some parts of Oromia region. These regions demonstrate significant clustering of minimum MMF values, indicating a lower prevalence of MMF compared to other parts of the country. The presence of these cold spots suggests a regional concentration of practices or factors that contribute to lower MMF rates in these areas.

Overall, the Hot Spot Analysis using Getis-Ord Gi* provides valuable insights into the spatial distribution of MMF across Ethiopia, highlighting specific regions with significantly higher or lower prevalence, and identifying areas with no significant spatial patterns (Fig 3).

**Table 1. Sociodemographic Characteristics of Mixed Milk Feeding Practices Among Infants Aged 0–5 Months in Ethiopia, EDHS 2019 (n = 524).**

| Variable | Categories | Weighted proportion of Feeding status | |
| --- | --- | --- | --- |
| | | Non_MMF | MMF |
| Sex of the child | Male | 23 (4.33%) | 221 (42.2%) |
| | Female | 30 (5.79%) | 249 (47.8%) |
| Age of the mother | 15–24 | 151 (91.6%) | 14 (8.4) |
| | 25–34 | 225 (87.4%) | 33 (12.6%) |
| | 35–49 | 95 (94%) | 7 (6.4) |
| Place of residence | Urban | 98 (81.9%) | 22 (18.1%) |
| | Rural | 372 (92.2%) | 31 (7.8) |
| Educational status | No education | 242 (93.2%) | 18 (6.8%) |
| | Primary education | 168 (88.6%) | 22 (11.4%) |
| | Secondary education and above | 61 (81.7%) | 14 (18.3) |
| Marital status | Married | 441 (89.8%) | 50 (10.2%) |
| | Not married | 30 (91.7%) | 3 (8.3%) |
| Wealth index level | Poor | 117 (89.4%) | 14 (3.2%) |
| | Middle | 220 (96.8%) | 7 (3.2%) |
| | Rich | 134 (80.8%) | 32 (19.2%) |
| Sex of child | Male | 221 (90.7%) | 23 (9.3%) |
| | Female | 250 (89.9%) | 30 (10.8%) |
| Age of infant | 0-3 months | 293 (93.2%) | 21 (6.8%) |
| | 4-5 months | 177 (89.9%) | 3 (15.1%) |
| Regions | Tigray | 41 (93.7%) | 3 (6.3) |
| | Afar | 6 (69.6%) | 3 (30.4%) |
| | Amhara | 84 (91.1%) | 8 (8.9%) |
| | Oromia | 204 (90.8%) | 20 (8.9%) |
| | Somali | 23 (72.2%) | 8 (25.8%) |
| | Benshangul Gumz | 6 (85.7%) | 1 (14.3%) |
| | SNNPR | 90 (94.5%) | 5 (5.5%) |
| | Gambela | 3 (88.1%) | 1 (11.9) |
| | Harari | 2 (66.7%) | 1 (33.3) |
| | Addis Abeba | 10 (77.2%) | 3 (22.8%) |
| | Dire Dawa | 2 (66.3%) | 1 (33.7%) |

## Mixed milk feeding practice cluster and outlier (Anselin local Moran's I) analysis

In this study, cluster and outlier analysis of MMF was conducted in Ethiopia using Anselin Local Moran's I spatial statistics to determine the presence of significant high or low values that are surrounded by similarly high or low values (clusters) or by contrasting values (outliers). The map shows regions categorized as 'Not Significant' (yellow dots), 'High-High Cluster' (red dots), 'High-Low Outlier' (purple dots), 'Low-High Outlier' (cyan dots), and 'Low-Low Cluster' (blue dots). The high-high significant cluster areas were observed in the regions of eastern and northern Somali, Dire Dawa and southern and eastern Afar. These regions are characterized by a high prevalence of MMF surrounded by other areas with similarly high rates.

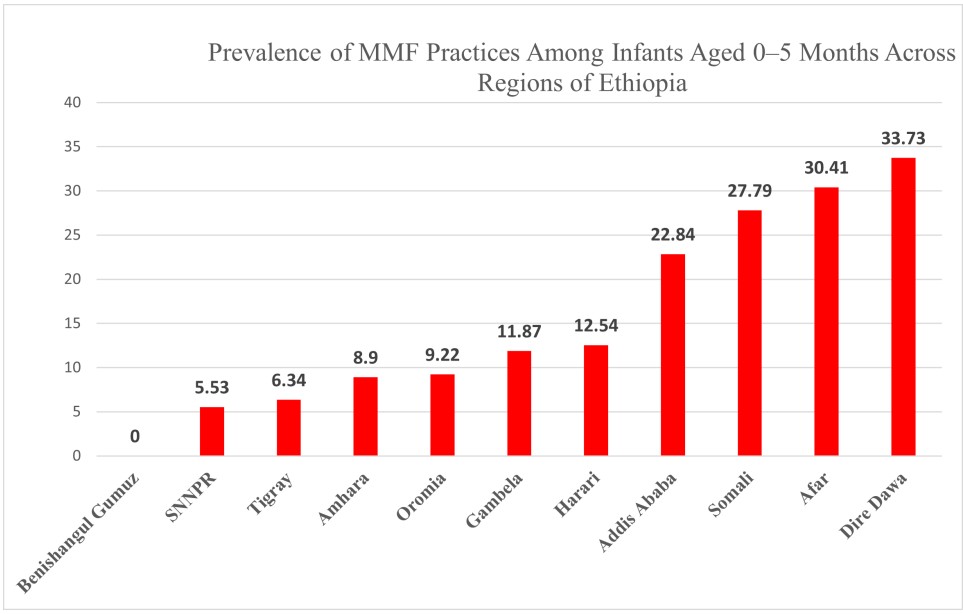

**Fig 1. Prevalence of mixed milk feeding among infants less than 6 months old across different regions of Ethiopia, Mini-EDHS 2019.**

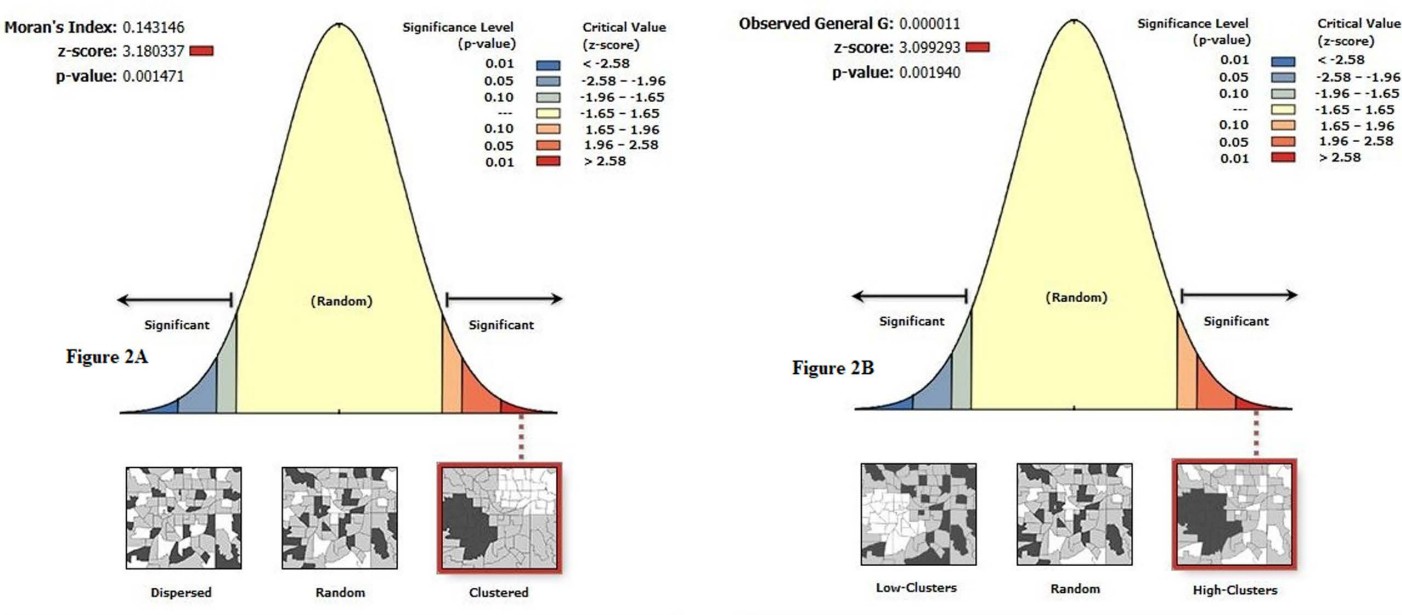

**Fig 2. Spatial autocorrelation analysis result of clustered patterns ( Fig 2A) and Spatial autocorrelation indicating high clusters (Fig 2B) of mixed milk feeding among infants less than 6 months old in Ethiopia, Mini-EDHS 2019.**

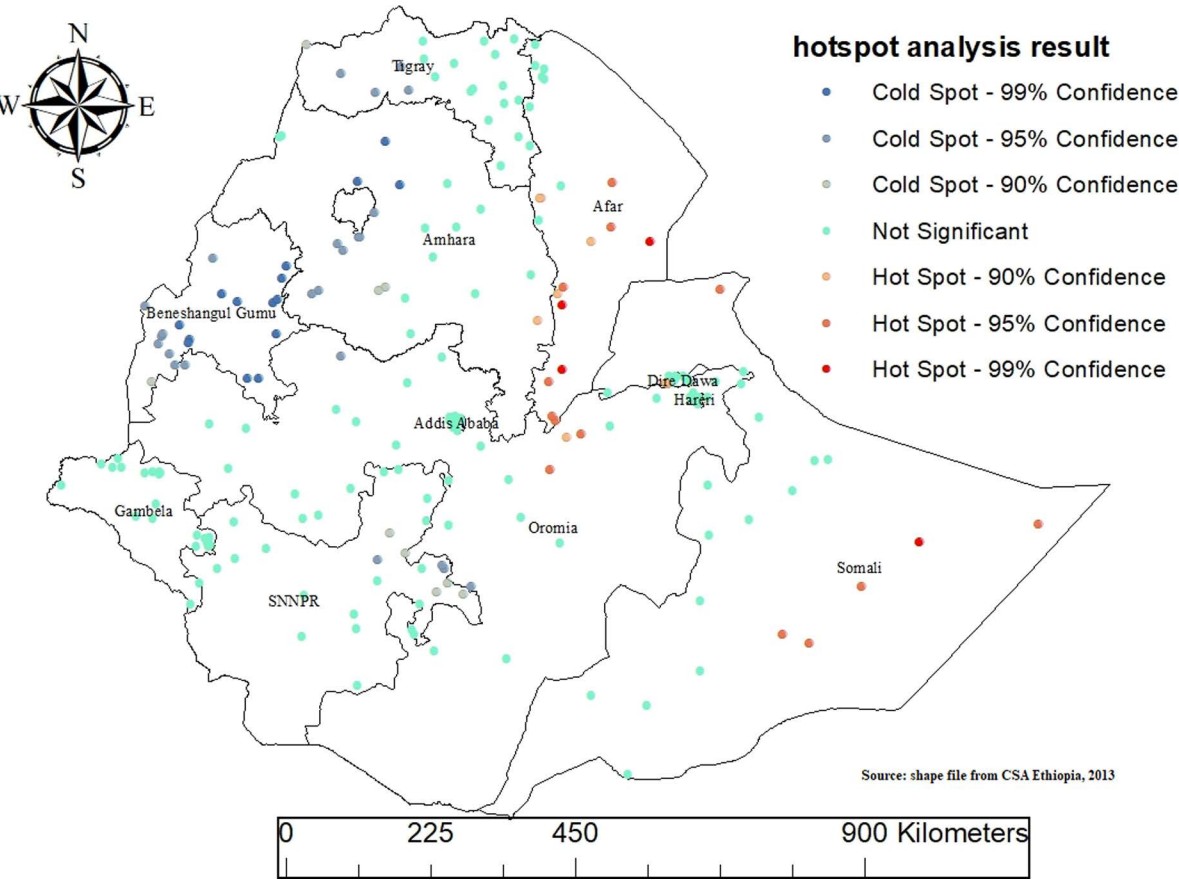

**Fig 3. Hot spots analysis result mixed milk feeding practice in Ethiopia, EDHS 2019.**

Conversely, the low-low significant clusters were observed in eastern SNNPR, northern Oromia, Benishangul-Gumuz, Tigray, Amhara and some parts of Dire Dawa. These areas exhibit a low prevalence of MMF surrounded by other areas with similarly low rates.

Overall, the high-high clusters denote regions with a high prevalence of MMF surrounded by high rates. High-low signifies areas with a high prevalence of MMF amidst low rates, while low-high indicates locales with a low prevalence of MMF surrounded by high rates. Lastly, low-low identifies areas with a low prevalence of MMF surrounded by low rates. This analysis provides a detailed spatial understanding of MMF practices across Ethiopia, highlighting regions for targeted interventions and policy efforts (Fig 4).

## Spatial interpolation

Spatial interpolation, a method crucial for estimating values in areas lacking direct measurements, was utilized to estimate unsampled areas using sampled data points. Empirical Bayesian Kriging (EBK) was specifically employed to account for errors in estimating the semivariogram model, essential for characterizing spatial dependence. Empirical Bayesian Kriging integrates data variability and spatial correlation to enhance prediction accuracy, making it particularly effective where data points are sparse or irregularly distributed. This approach ensures robust spatial predictions while addressing uncertainties inherent in the interpolation process, valuable for applications in environmental management and urban planning (Fig 5).

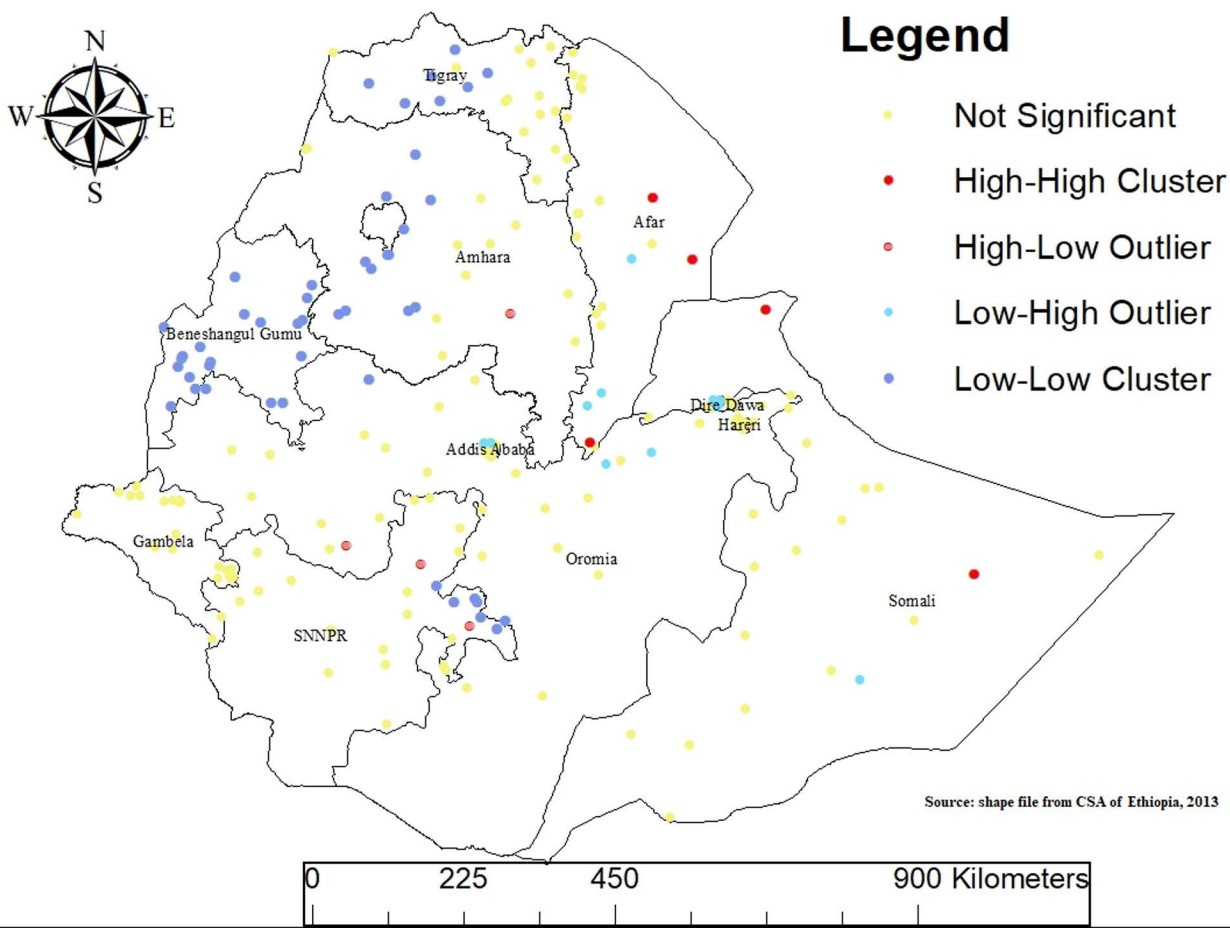

**Fig 4.** Fig 3 **Outlier and Hotspot Analysis of Mixed Milk Feeding (MMF) in Ethiopia, EDHS 2019.**

## SaTScan cluster analysis of mixed milk feeding among mothers with infants aged 0-5 months in Ethiopia

The SaTScan cluster analysis identified significant spatial clusters of MMF practices among mothers with infants aged 0-5 months in Ethiopia. A purely spatial analysis was conducted using the Bernoulli model to identify clusters with high or low rates. The SaTScan analysis identified a significant spatial cluster of MMF practices among infants aged 0-5 months in Ethiopia.

This cluster was located in Eastern Ethiopia, covering regions such as Somali, Oromia, Dire Dawa, Hareri, Addis Abeba, and Afar, with coordinates 8.418319 N latitude and 43.841669 E longitude, and a radius of 587.71 km. Infants within this cluster exhibited a Relative Risk of 4.8 (P-value ≤ 0.001) and a Log-ikelihood Ratio of 16.5, indicating they were more than four times as likely to receive MMF compared to infants outside this region. These findings underscore localized variations in infant feeding practices, highlighting potential socio-cultural influences specific to Eastern Ethiopia (Fig 6).

## Spatial regression result

**OLS analysis result.** The OLS model's results showed that about 6.2% of the MMF variation was explained by the independent variable, indicating a good fit (adjusted

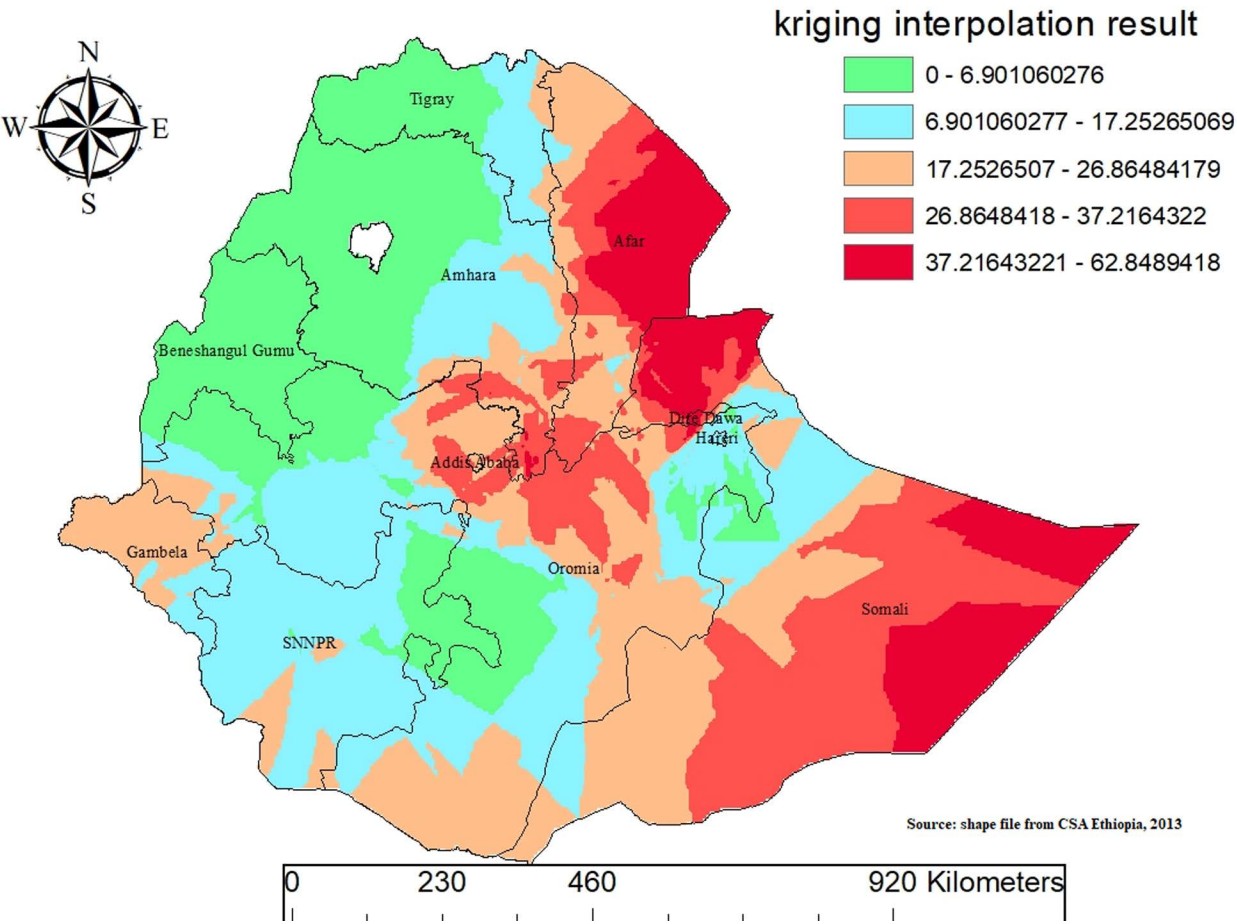

**Fig 5. Kriging interpolation of mixed milk feeding in Ethiopia, EDHS 2019.**

R2 = 0.062). The variance inflation factor value for the OLS model in this study was less than seven, suggesting no multicollinearity. The joint Wald statistics were statistically significant (p < 0.001), indicating that the overall model was significant (Table 2). The Jarque-Bera statistics were not significant (p = 0.08), implying that the model residuals were normally distributed. The Koenker statistics test was found to be statistically significant and showed a nonstationary relationship between the predictor variables and EBF (p < 0.001). This suggests the need to consider the relationship between the predictor variables and MMF by obtaining a local coefficient for every explanatory variable. The proportion of middle wealth index and baby without a postnatal checkup within two months were predictors of mixed milk feeding.

## GWR results

The OLS analysis predicted hotspot areas but assumed a stationary relationship between predictor and outcome variables. This assumption was violated in the study (Koenker statistics, p < 0.001), indicating the need for a GWR model to account for local variations in predictor effects. Running a GWR model revealed significant improvement by identifying local variations in predictor effects across the study area. The Akaike's information criterion value decreased from 108.15 in the OLS model to 89.24 in the GWR model. In addition, the adjusted

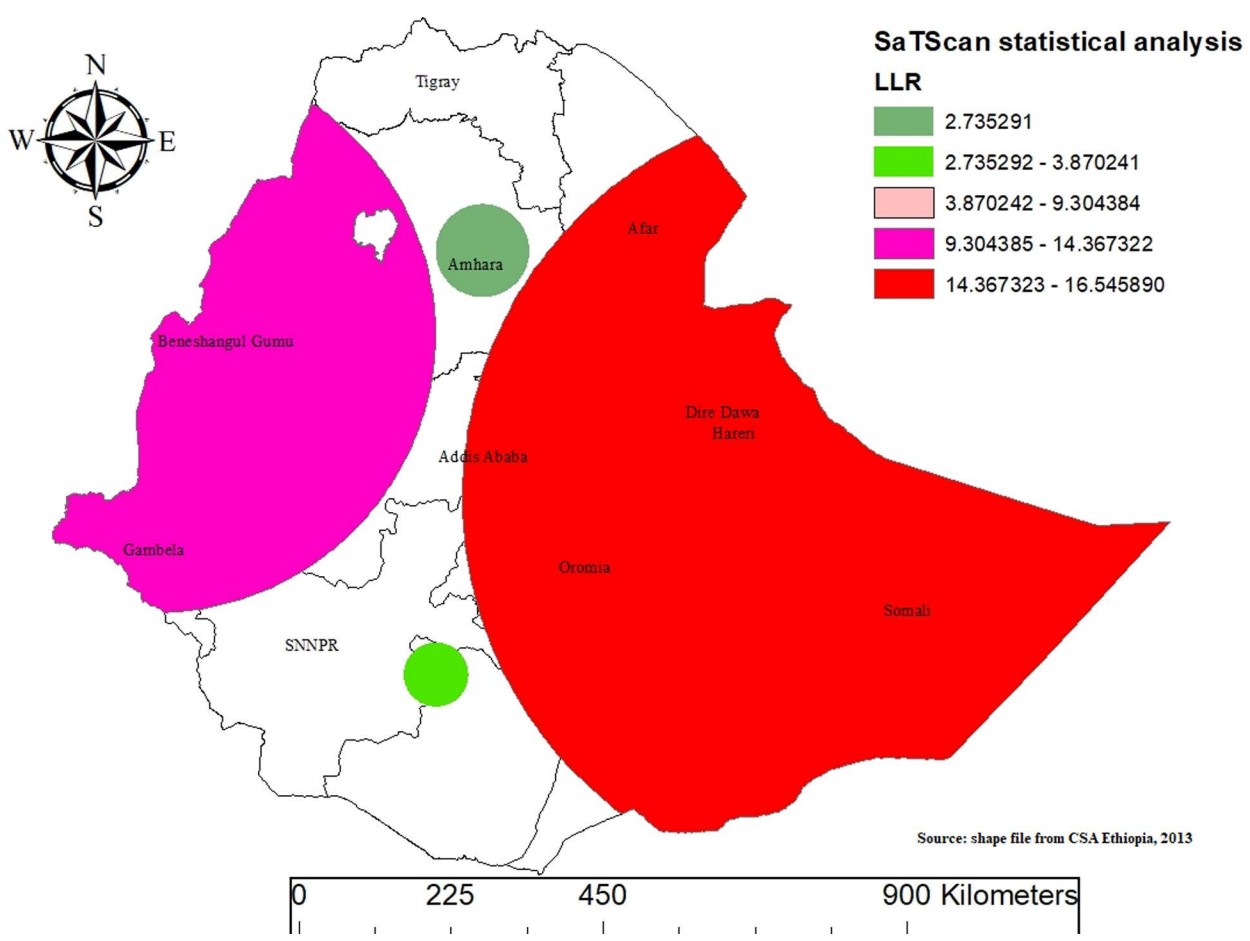

**Fig 6. Spatiotemporal patterns of mixed milk feeding in Ethiopia.**

R2 obtained from OLS increased from 6.2% to 12.38%, which implies that GWR analysis improved the model's ability to predict MMF better than the OLS model (Table 3).

We examined the spatial patterns in the residuals to validate our GWR analysis results. The Moran's I statistic for the residuals from the GWR model analyzing MMF practices among infants aged 0–5 months yielded a Moran's I value of 0.023 with a p-value of 0.550. This indicates that the residuals are randomly distributed without significant spatial clustering or dispersion. These results suggest that the GWR model effectively captured the spatial patterns in MMF practices and that the variables included in the model adequately explained the spatial dependencies (Fig 7).

The findings from the Geographically Weighted Regression (GWR) analysis indicate that MMF practices in Ethiopia are significantly influenced by variables such as households belonging to the middle wealth index and babies without a postnatal checkup. These factors exhibit spatial variability in their impact across diverse regions of the country.

The relationship between babies without a postnatal checkup and MMF practices varies across different regions, showing both positive and negative associations. In most parts of Ethiopia, babies without a postnatal checkup are positively associated with MMF practices. This positive association is especially prominent in regions such as Somali, Afar, Tigray, Amhara, Dire Dawa, Harari, most parts of Oromia, and some parts of the SNNPR (Southern

**Table 2. Summary of OLS results mixed milk feeding practice in Ethiopia, EDHS 2019.**

| Variable | Coefficient [a] | StdError | t-Statistic | Probability [b] | Robust_SE | Robust_t | Robust_Pr [b] | VIF [c] |
|---|---|---|---|---|---|---|---|---|
| Intercept | 0.14 | 0.12 | 1.24 | 0.22 | 0.10 | 1.39 | 0.17 | -------- |
| Rural residence | -0.04 | 0.06 | -0.63 | 0.53 | 0.06 | -0.65 | 0.52 | 2.10 |
| Women with no education | -0.05 | 0.08 | -0.66 | 0.51 | 0.08 | -0.66 | 0.51 | 2.96 |
| Women with primary education | -0.09 | 0.07 | -1.29 | 0.20 | 0.07 | -1.25 | 0.21 | 2.22 |
| Delivered by CS | 0.12 | 0.09 | 1.29 | 0.20 | 0.12 | 1.04 | 0.30 | 1.30 |
| Home delivery | 0.07 | 0.07 | 1.01 | 0.31 | 0.05 | 1.25 | 0.21 | 2.21 |
| Baby without a postnatal checkup | 0.16 | 0.08 | 2.06 | <0.05* | 0.07 | 2.35 | <0.01* | 1.09 |
| Middle wealth index | -0.15 | 0.06 | -2.43 | <0.01* | 0.05 | -2.75 | <0.01* | 1.71 |
| Rich wealth index | 0.01 | 0.07 | 0.12 | 0.90 | 0.06 | 0.14 | 0.89 | 3.16 |
| Breastfeeding counselling | -0.05 | 0.06 | -0.82 | 0.41 | 0.06 | -0.77 | 0.44 | 1.57 |
| **OLS Diagnostics** | | | | | | | | |
| Number of Observations: | 238 | | | (AICc) [d]: | | | 108.15 | |
| Multiple R-Squared [d]: | 0.098 | | | Adjusted R-Squared [d]: | | | 0.062 | |
| Joint F-Statistic [e]: | 2.75 | | | Prob(>F), (9,228) degrees of freedom: | | | <0.01* | |
| Joint Wald Statistic [e]: | 25.07 | | | Prob(>chi-squared), (9) degrees of freedom: | | | <0.01* | |
| Koenker (BP) Statistic [f]: | 21.19 | | | Prob(>chi-squared), (9) degrees of freedom: | | | <0.01* | |
| Jarque-Bera Statistic [g]: | 121.56 | | | Prob(>chi-squared), (2) degrees of freedom: | | | 0.08 | |

AICc, Akaike's information criterion; EDHS, Ethiopian Demographic and Health Survey; GWR, geographically weighted regression; OLS, ordinary least squares.

**Table 3. Geographic weighted regression (GWR) model for mixed milk practices in Ethiopia, EDHS 2019.**

| Explanatory variable | Households belonging to middle wealth index and baby postnatal check |
|---|---|
| Residual R squares | 18.10 |
| Effective number | 16.17 |
| Stigma | 0.28 |
| AICc | 89.24 |
| Multiple R2 | 17.99 |
| Adjusted R2 | 12.38 |
| **Model comparison** | |

| Parameters | OLS model | GWR model |
|---|---|---|
| AICc | 108.15 | 89.24 |
| Multiple R2 | 9.8 | 17.99 |
| Adjusted R2 | 6.2 | 12.38 |

Nations, Nationalities, and Peoples' Region). In these areas, the lack of postnatal checkups correlates with a higher likelihood of engaging in MMF practices. This is represented by the red points on the map (GWR coefficient: 0.220599 to 0.542988). Areas with a moderate positive association are indicated by the yellow points (GWR coefficient: 0.145158 to 0.220598).

Conversely, there are regions where the absence of a postnatal checkup is significantly negatively associated with MMF practices. Notable areas with this negative association include Gambela, Benishangul-Gumuz, some parts of Oromia, and parts of the SNNPR region. In these regions, babies without postnatal checkups are less likely to be involved in MMF practices. The negative association is indicated by the green points (GWR coefficient: -0.261378

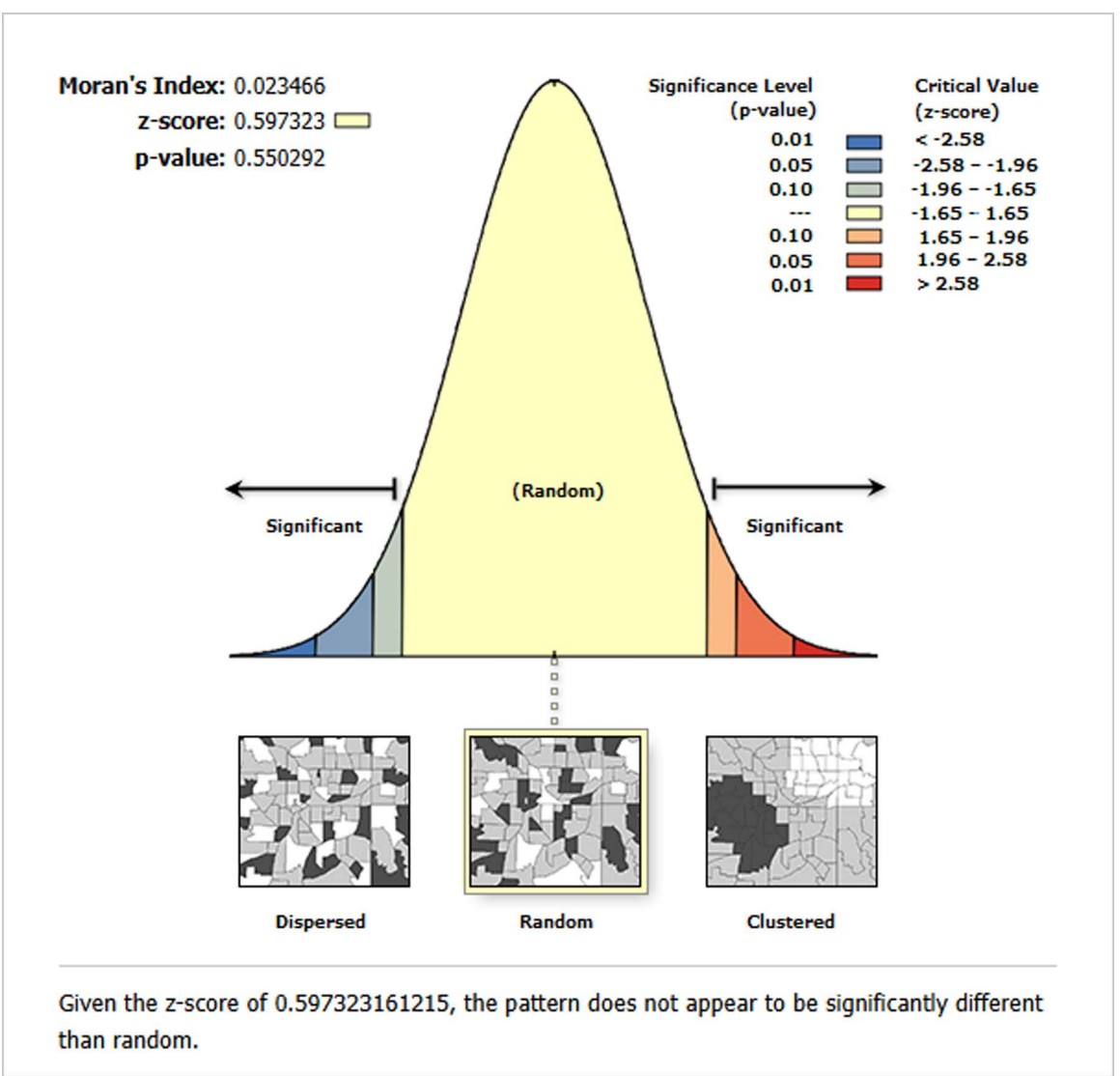

**Fig 7.  Moran's I Statistic for Residuals of the GWR Model on MMF practices among infant aged 0-5 months in Ethiopia, 2019.**

to -0.085330) and light green points (GWR coefficient: -0.085329 to -0.052690). Areas with a moderate negative association are indicated by the light-yellow points (GWR coefficient: 0.052691 to 0.145157). The GWR (Geographically Weighted Regression) model provides a detailed spatial analysis, highlighting the regional variations in the impact of postnatal checkups on MMF practices. The findings underscore the importance of postnatal healthcare services in influencing infant feeding behaviors and emphasize the need for targeted interventions that consider regional disparities across Ethiopia (Fig 8).

Households in the middle wealth index are statistically significant predictors of MMF (Mixed Milk Feeding) practices across Ethiopia. There is a negative association between middle-wealth index households and MMF practices in most regions. Specifically, areas with stronger negative associations are indicated by green points (-0.256480 to -0.153245). Moderate negative associations are represented by yellow to orange points (-0.153245 to -0.076699), while weaker or slightly positive associations are indicated by red points (-0.076699 to 0.151018). Notable regions with a pronounced negative association include parts of the

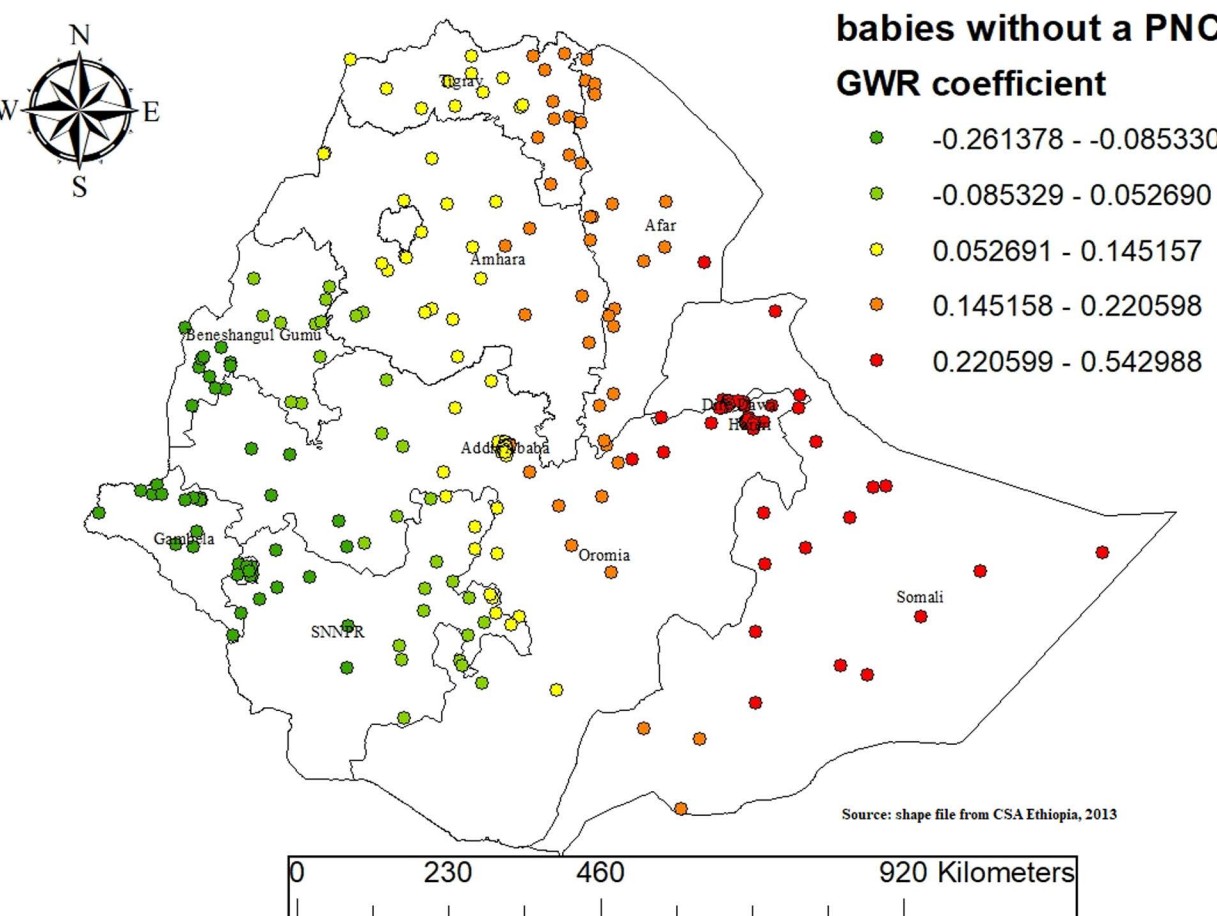

**Fig 8.  GWR analysis result of Baby without a postnatal checkup coefficient for predicting mixed milk feeding in Ethiopia, EDHS 2019.**

northern and central areas such as Amhara and Oromia. Conversely, some areas in the western and southern regions exhibit a less negative or slightly positive association, indicating regional variations in the influence of the middle wealth index on MMF practices (Fig 9).

## Discussion

This study represents the first investigation in Ethiopia to explore the spatial distribution and determinants of MMF among mothers with infants aged 0-5 months. The prevalence of MMF was found to be 10.12% (95% CI 7.8, 13.01), with a notable hotspot identified in the eastern regions of Ethiopia, including Somali, Afar, and Dire Dawa. Conversely, significant cold spot areas were observed in much of Amhara, Tigray, Benishangul Gumuz, SNNPR, and parts of Oromia. Geographically Weighted Regression analysis underscored that MMF practices in Ethiopia are influenced by factors such as household wealth (middle wealth index) and the absence of postnatal checkups for infants. These findings provide valuable insights into regional disparities and key determinants shaping MMF practices, highlighting the need for targeted interventions to improve infant feeding practices across Ethiopia.

The present study identified the prevalence of MMF at 10.12%, which significantly differs from the results reported in a 2020 systematic review and meta-analysis [24]. This notable difference suggests potential regional variations in MMF practices influenced by diverse cultural

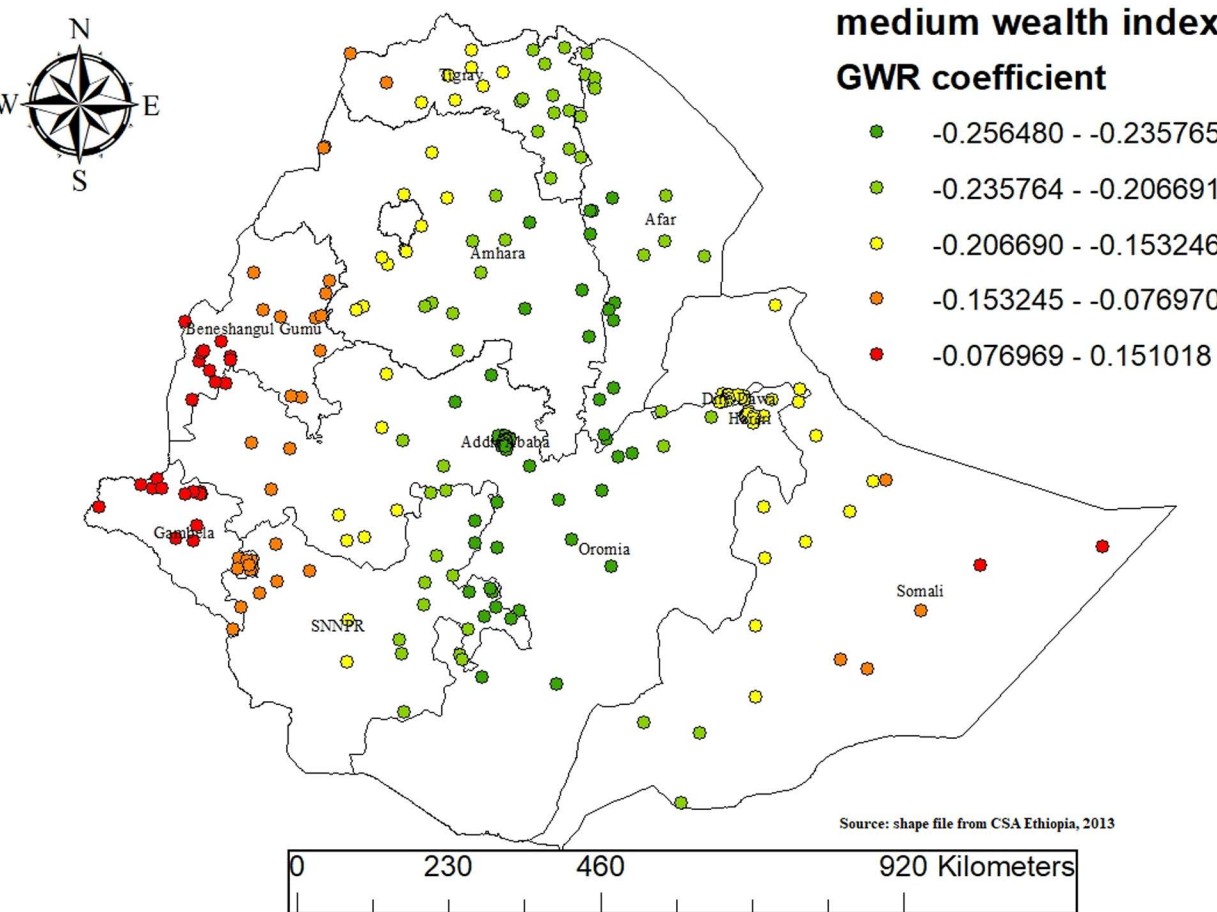

**Fig 9. GWR analysis result of households in the middle wealth index coefficient for predicting mixed milk feeding in Ethiopia, EDHS 2019.**

norms, healthcare systems, and socioeconomic factors. Factors contributing to Ethiopia's lower prevalence could include maternal employment patterns, or traditional feeding practices specific to the region. Moreover, the high cost and limited availability of formula products in Ethiopia are substantial deterrents for mothers considering MMF. In Ethiopia, formula products are often prohibitively expensive and inaccessible [47], making them less practical as many communities lack resources or infrastructure to support MMF.

Hotspot areas of MMF in eastern Ethiopia, particularly in the Somali, Afar, and Dire Dawa can be attributed to a unique combination of cultural norms, widespread cattle herding providing abundant animal milk, and the nomadic lifestyle prevalent among inhabitants. These regions rely heavily on cattle for sustenance [48,49], ensuring a readily available milk supply that facilitates MMF practices. Limited healthcare access and the pastoralist lifestyle pose challenges in delivering adequate health services, hindering efforts to promote exclusive breastfeeding and contributing to disparities in healthcare distribution and maternal awareness of its benefits [28]. Health services, including infant feeding counseling, are rarely available in these temporary grazing areas, which often lack the necessary infrastructure and resources to support such services [50]. In Eastern Ethiopia, pastoralist communities such as those in Afar and Somali regions maintain strong adherence to traditional cultural and religious values [51]. This practice rooted in deep-seated traditions, considers animal milk a culturally accepted and

practical substitute when breastfeeding is perceived as insufficient, aligning closely with their lifestyle and belief systems.

In most parts of Ethiopia, babies without a postnatal checkup are positively associated with MMF practices. This positive association is especially prominent in hot spots such as Somali, Afar, Tigray, Amhara, Dire Dawa, Harari, most parts of Oromia, and some parts of the SNNPR. In these areas, the lack of postnatal checkups correlates with a higher likelihood of engaging in MMF practices. Possible justification for this could be that when these checkups are missed, mothers often lack crucial information and support that could encourage exclusive breastfeeding. During follow-up mothers/caregivers will received information from health care providers on duration, frequency, compositions, and benefits of EBF [52].

The current study identified a negative association between middle-wealth index households and MMF practices in all regions of Ethiopia. This trend may be attributed to several factors. Middle-wealth households often have better access to health education and resources promoting exclusive breastfeeding compared to lower-wealth households [53], which can lead to higher adherence to recommended feeding practices. Additionally, these households might have better access to health-care facilities and postnatal checkups, where mothers receive guidance on the benefits of exclusive breastfeeding. It can also be suggested that mothers from the poorest wealth index group might perceive themselves as producing inadequate breast milk to satisfy their infant's demand, leading them to initiate additional foods [54]. The economic stability of middle-wealth households also reduces the need for mixed milk feeding, as they can afford to follow health recommendations more closely.

However, while middle-wealth households generally adhere to exclusive breastfeeding due to better access to health education and resources, high-wealth households might be more likely to use infant formula in addition to breastfeeding because they can afford it and may perceive it as a convenient or premium option [55]. Wealthier families often use formula or non-human milk in addition to breastfeeding for infants under six months old due to their greater financial resources, which allow them to purchase these options perceived as convenient and premium. This practice offers flexibility for busy lifestyles by enabling other caregivers to feed infants, and it may be influenced by marketing and societal trends that promote supplemental feeding as a modern alternative to exclusive breastfeeding [56].

The study utilized nationally representative data to investigate the spatial heterogeneity factors associated with MMF providing valuable insights for intervention strategies. Analysis adjustments, such as weighting and accounting for sample design, were applied to ensure representation of the national population. However, limitations exist within this study, such as adjustments to the geographical locations of enumeration areas due to data privacy concerns, which could potentially affect the accuracy of spatial analysis in clusters. This study provides valuable insights into the spatial distribution and determinants of MMF in Ethiopia, highlighting regional disparities and key factors influencing feeding practices. The findings suggest that targeted interventions addressing regional needs, improving access to healthcare services, and enhancing maternal education on exclusive breastfeeding are crucial. Future research should consider longitudinal studies to establish causality and explore additional factors such as maternal health status, knowledge, and attitudes towards MMF. Additionally, efforts to address the high cost and limited availability of formula products could help reduce the reliance on MMF and promote exclusive breastfeeding practices. Overall, this research underscores the need for tailored strategies to address the diverse factors affecting MMF practices across different regions of Ethiopia and enhance infant feeding practices nationally.

## Conclusion

The study revealed substantial regional variations in MMF practices among Ethiopian women, with significant differences in prevalence and distribution across the country. Somali, Dire Dawa, and the Afar region were identified as MMF hotspots, while Amhara, Tigray,

Benishangul Gumuz, SNNPR, and parts of Oromia were cold spots. MMF practices were influenced by household wealth, especially within the middle wealth index, and the absence of postnatal checkups.

Interventions are needed to promote exclusive breastfeeding in high MMF prevalence areas. Educational initiatives about breastfeeding benefits should be prioritized within maternal health services. Cross-sector collaboration among health, education, and media sectors is essential for consistent messaging. Increasing the accessibility and affordability of postnatal healthcare services through mobile clinics and community health workers can ensure all mothers receive necessary breastfeeding support. Integrating MMF counseling into routine healthcare services can provide mothers with essential information and support. Further research is needed to tailor interventions to local cultural factors, addressing undernutrition and reducing child morbidity and mortality associated with suboptimal breastfeeding practices.

## Acknowledgments

We acknowledge The DHS Program for granting us access to utilize the Ethiopia Demographic and Health Survey (EDHS) data for our analysis.

## Author contributions

**Conceptualization:** Mekuriaw Nibret Aweke, Muluken Chanie Agimas, Moges Tadesse Abebe, Tigabu Kidie Tesfie, Meron Asmamaw Alemayehu, Worku Necho Asferie.

**Data curation:** Gebrie Getu Alemu.

**Formal analysis:** Mekuriaw Nibret Aweke, Gebrie Getu Alemu, Worku Necho Asferie.

**Investigation:** Mekuriaw Nibret Aweke.

**Methodology:** Mekuriaw Nibret Aweke, Meron Asmamaw Alemayehu, Werkneh Melkie Tilahun, Gebrie Getu Alemu, Worku Necho Asferie.

**Writing – original draft:** Mekuriaw Nibret Aweke, Meron Asmamaw Alemayehu.

**Writing – review & editing:** Mekuriaw Nibret Aweke, Muluken Chanie Agimas, Meron Asmamaw Alemayehu, Werkneh Melkie Tilahun, Gebrie Getu Alemu, Worku Necho Asferie.

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
