## [Decision Letter · Decision Letter 0]

10 Sep 2024

PONE-D-24-28246Spatial distribution of mixed milk feeding and its determinants among mothers of infants aged under 6 months in Ethiopia: spatial and geographical weighted regression analysis.PLOS ONE

Dear Dr. Aweke,

Thank you for submitting your manuscript to PLOS ONE. After careful consideration, we feel that it has merit but does not fully meet PLOS ONE’s publication criteria as it currently stands. Therefore, we invite you to submit a revised version of the manuscript that addresses the points raised during the review process.

Please note that we have only been able to secure a single reviewer to assess your manuscript. We are issuing a decision on your manuscript at this point to prevent further delays in the evaluation of your manuscript. Please be aware that the editor who handles your revised manuscript might find it necessary to invite additional reviewers to assess this work once the revised manuscript is submitted. However, we will aim to proceed on the basis of this single review if possible. 

We look forward to receiving your revised manuscript.

Kind regards,

Jennifer Tucker, PhD

Staff Editor

PLOS ONE

Journal Requirements: When submitting your revision, we need you to address these additional requirements. 1. Please ensure that your manuscript meets PLOS ONE's style requirements, including those for file naming. The PLOS ONE style templates can be found at https://journals.plos.org/plosone/s/file?id=wjVg/PLOSOne_formatting_sample_main_body.pdf and https://journals.plos.org/plosone/s/file?id=ba62/PLOSOne_formatting_sample_title_authors_affiliations.pdf 2. We noticed you have some minor occurrence of overlapping text with the following previous publication(s), which needs to be addressed: -https://bmjpaedsopen.bmj.com/content/8/Suppl_2/e002573-https://www.scielo.br/j/rbsmi/a/MRmNFWnRbmv8XZ7BnQd4D8P/?lang=en In your revision ensure you cite all your sources (including your own works), and quote or rephrase any duplicated text outside the methods section. Further consideration is dependent on these concerns being addressed. 3. We note that your Data Availability Statement is currently as follows: All relevant data are within the manuscript and its Supporting Information files Please confirm at this time whether or not your submission contains all raw data required to replicate the results of your study. Authors must share the “minimal data set” for their submission. PLOS defines the minimal data set to consist of the data required to replicate all study findings reported in the article, as well as related metadata and methods (https://journals.plos.org/plosone/s/data-availability#loc-minimal-data-set-definition). For example, authors should submit the following data: - The values behind the means, standard deviations and other measures reported;- The values used to build graphs;- The points extracted from images for analysis. Authors do not need to submit their entire data set if only a portion of the data was used in the reported study. If your submission does not contain these data, please either upload them as Supporting Information files or deposit them to a stable, public repository and provide us with the relevant URLs, DOIs, or accession numbers. For a list of recommended repositories, please see https://journals.plos.org/plosone/s/recommended-repositories. If there are ethical or legal restrictions on sharing a de-identified data set, please explain them in detail (e.g., data contain potentially sensitive information, data are owned by a third-party organization, etc.) and who has imposed them (e.g., an ethics committee). Please also provide contact information for a data access committee, ethics committee, or other institutional body to which data requests may be sent. If data are owned by a third party, please indicate how others may request data access. 4. Please amend the manuscript submission data (via Edit Submission) to include author Dr. Tigabu Kidie Tesfie. 5. We note you have included a table to which you do not refer in the text of your manuscript. Please ensure that you refer to Table 1 in your text; if accepted, production will need this reference to link the reader to the Table.

Reviewers' comments:

Reviewer's Responses to Questions

**Comments to the Author**

1. Is the manuscript technically sound, and do the data support the conclusions?

Reviewer #1: Yes

2. Has the statistical analysis been performed appropriately and rigorously? 

Reviewer #1: Yes

3. Have the authors made all data underlying the findings in their manuscript fully available?

Reviewer #1: Yes

4. Is the manuscript presented in an intelligible fashion and written in standard English?

Reviewer #1: Yes

5. Review Comments to the Author

Reviewer #1: Title: “Spatial distribution of mixed milk feeding and its determinants among mothers of infants aged under 6 months in Ethiopia: spatial and geographical weighted regression analysis”

Here is an in-depth report addressing each of the checklist items and providing specific sections where issues have been identified:

Title and Abstract:

1. (a) The title is well crafted.

(b) In the ABSTRACT:

• Method Section: You need to clearly describe the total weighted sample of infants used in your study.

• Result Section: Include the weighted prevalence of Mixed Milk Feeding (MMF) in Ethiopia. Additionally, it's recommended to include numerical results such as the global Moran's I with p-value, the percentage of the maximum cluster size per population, etc.

• Keywords: Why are the keywords missing? Please include them in your abstract and arrange them according to scientific standards.

• Overall: The abstract does not effectively convey what was done in the study, so revisions are needed for clarity and completeness.

INTRODUCTION:

2. The introduction section presents the following issues:

• In paragraph 5: “Studies have shown that breastfeeding improves the emotional and psychological connection between mother and infant.” What are the studies? Lacks a relevant litrature.

• In paragraph 9: “To this day, various research has been carried out in Ethiopia to evaluate the exclusive breast feeding among children, including the bottle feeding.” Please put the relevance various research reference to establish the rationale.

• Overall, the introduction is well written.

METHODS AND MATERIALS:

3. Data source: The description of the setting, and locations is correctly provided. I think this is a secondary data analysis, the absence of eligibility criteria and participant selection details is acceptable so I didn’t see such thing in detail.

4. Sampling procedure and population: Do you believe that 'all living infants under 6 months of age' amounts to only 524 (based on the 2019 EMDHS)? If not, how was the sample of 524 infants selected? Please provide a clear description of the selection procedure.

5. Data management and analysis: Please put the version of STATA. “We remove missing values from our analysis by using the STATA drop command in combination with a logical or conditional statement.” But you mentioned “Missing data were managed based on the DHS guideline” under data source section. How do you relate those two sentences? Do you believe that “Removing” is considered as the management of missing data analysis? If not, mange your missing data based on imputation or other handling method.

6. Please thoroughly document all methods under the 'Spatial Scan Statistics' and 'Spatial Regression' sections, including relevant supporting references.

RESULTS:

7. Please format the table title clearly (e.g., 'Problem, Population, Area, 2024, n=') and ensure the tables are cited throughout the document with cross-references.

8. Prevalence of mixed milk feeding in Ethiopia:” In this study, the overall prevalence of MMF was 10.12% (95% CI 7.8, 13.01), with regional variations observed across Ethiopia”. The results were not displayed in any tables or graphs. Please provide the prevalence data in either a table or a graph, and reference any tables used.

DISCUSSION:

9. The 'Discussion' section can be generally divided into 3 separate paragraphs. 1) Introductory paragraph/rationale of the study, 2) Intermediate paragraphs/compare and contrast with the most recent and relevant literature, 3) Concluding paragraph/indicating future directions. Then address such things as well.

10. While the manuscript mentions the study utilized nationally representative data as a strength indicating there is adequate power to detect the true effect of exposures. However, it does not provide a power calculation to support this claim.

Recommendation

This comprehensive report outlines the issues identified throughout the manuscript, with specific sections and requires substantial minor revisions to address these concerns and enhance the clarity, relevance, and presentation of results. After making the necessary minor revisions with re-write of manuscript, a re-evaluation is recommended for considering publication.

6. PLOS authors have the option to publish the peer review history of their article (what does this mean? ). If published, this will include your full peer review and any attached files.

**Do you want your identity to be public for this peer review?** For information about this choice, including consent withdrawal, please see our Privacy Policy .

Reviewer #1: **Yes: ** Abdu Hailu Shibeshi

---

## [Author Response · Author response to Decision Letter 0]

30 Sep 2024

PLOS ONE Editorial Office Date 19 September 2024

Dear Editor,

Subject: Resubmission of Revised Manuscript PONE-D-24-28246

I am pleased to resubmit the revised version of our manuscript titled "Spatial distribution of mixed milk feeding and its determinants among mothers of infants aged under 6 months in Ethiopia: spatial and geographical weighted regression analysis" (Manuscript ID: PONE-D-24-28246). We have carefully addressed the comments and suggestions provided by the reviewers and editorial team.

In the accompanying response document, we have detailed how each point was addressed, and we have made appropriate revisions to the manuscript. We believe these changes have improved the quality of our work, and we are hopeful that the revised version now meets PLOS ONE's publication standards.

Thank you for the opportunity to revise and resubmit our manuscript. We look forward to your feedback and hope for a favorable decision.

Sincerely,

Mekuriaw Nibret Aweke

Correspondent Author

Comments from the Academic Editor and Responses

Authors’ Response:

Dear editor,

Thank you for the opportunity to submit our manuscript. We have corrected the document to fulfill PLOS ONE publication requirements.

2. We noticed you have some minor occurrence of overlapping text with the following previous publication(s), which needs to be addressed.

Authors’ Response:

Dear editor,

Thank you for your feedback. We appreciate your feedback regarding the overlap with previous publications. We have carefully reviewed the sections identified and have made the revisions and correct citations.

3. Please confirm at this time whether or not your submission contains all raw data required to replicate the results of your study. Authors must share the “minimal data set” for their submission.

Authors’ Response:

Dear editor,

Thank you for your reminder regarding data availability. At this time, our submission does not include all raw data required to replicate the results of our study. The raw data used in our study, which is the ‘2019 Ethiopian Mini Demographic and Health Survey (miniDHS),’ can be accessed through the DHS Program (www.dhsprogram.com) as per their data access instructions. We mentioned this in the main document and under the data availability section.

4. Please amend the manuscript submission data (via Edit Submission) to include author Dr. Tigabu Kidie Tesfie.

Authors’ Response:

Dear editor,

Thank you for your notification. We have amended the manuscript submission to include Dr. Tigabu Kidie Tesfie as an author. The updated author list has been submitted through the journal’s submission system.

5. We note you have included a table to which you do not refer in the text of your manuscript. Please ensure that you refer to Table 1 in your text; if accepted, production will need this reference to link the reader to the Table.

Authors’ Response:

Dear editor,

Thank you for your feedback. We have updated the manuscript to include a reference to Table 1 in the text.

Comments from Reviewer #1 and Responses

(a) The title is well crafted.

(b) In the ABSTRACT: Method Section: You need to clearly describe the total weighted sample of infants used in your study.

Authors’ Response:

Dear reviewer,

Thank you for your feedback. We have updated the Method section of the ABSTRACT to clearly describe the total weighted sample of infants used in the study. The revised text now includes the specific number of infants in the weighted sample.

Reviewers’ Comments

Result Section: Include the weighted prevalence of Mixed Milk Feeding (MMF) in Ethiopia. Additionally, it's recommended to include numerical results such as the global Moran's I with p-value, the percentage of the maximum cluster size per population, etc.

Authors’ Response:

Dear reviewer,

Thank you for your suggestion and comments. We have included the weighted prevalence of Mixed Milk Feeding (MMF) with the numerical results of the global Moran's I with p-value and the maximum spatial cluster size per population in the abstract section of the methods and result parts. We appreciate your relevant feedback.

Reviewers’ Comments

Keywords: Why are the keywords missing? Please include them in your abstract and arrange them according to scientific standards.

Authors’ Response:

Dear reviewer,

Thank you for your feedback. We have included the keywords in the main document. Thank you again for your valuable comments.

Reviewers’ Comments

Overall: The abstract does not effectively convey what was done in the study, so revisions are needed for clarity and completeness.

Authors’ Response:

Dear Reviewer,

Thank you for your feedback. We have revised the abstract section accordingly. Thank you again for your unreserved comments and suggestions.

Reviewers’ Comments

In paragraph 5: “Studies have shown that breastfeeding improves the emotional and psychological connection between mother and infant.” What are the studies? Lacks relevant literature.

Authors’ Response:

Dear reviewer,

Thank you for your feedback. We have cited the relevant literature which showed “breastfeeding improves the emotional and psychological connection between mother and infant” in the main document. Thank you again for your valuable comments.

Reviewers’ Comments

In paragraph 9: “To this day, various research has been carried out in Ethiopia to evaluate the exclusive breastfeeding among children, including bottle feeding.” Please put the relevant various research references to establish the rationale.

Authors’ Response:

Dear reviewer,

Thank you for your suggestion. We have added the relevant studies’ citations to this statement. Thank you again for your relevant comments.

Reviewers’ Comments

Overall, the introduction is well written.

Authors’ Response:

Dear reviewer,

Thank you for your feedback! We are glad to hear that you found the introduction well written.

Reviewers’ Comments

Data source: The description of the setting, and locations is correctly provided. I think this is a secondary data analysis; the absence of eligibility criteria and participant selection details is acceptable so I didn’t see such things in detail.

Authors’ Response:

Dear reviewer,

Thank you for your feedback. We appreciate your understanding regarding the nature of our data since it is secondary data analysis. As noted, the absence of detailed eligibility criteria and participant selection information is a result of the use of pre-existing data, which was collected with its own criteria and protocols. Thank you again for your relevant suggestion and comments.

Reviewers’ Comments

Sampling procedure and population: Do you believe that 'all living infants under 6 months of age' amounts to only 524 (based on the 2019 EMDHS)? If not, how was the sample of 524 infants selected? Please provide a clear description of the selection procedure.

Authors’ Response:

Dear reviewer,

Thank you for your feedback. The 2019 Ethiopian Mini Demographic and Health Survey (EMDHS) initially included 618 infants under 6 months of age. For our study on MMF practices, we excluded infants who were not breastfeeding. This resulted in a final weighted sample of 524 infants who met the study criteria. We included this in the manuscript for further clarity of the sampling procedure.

Reviewers’ Comments

Data management and analysis: Please put the version of STATA. “We remove missing values from our analysis by using the STATA drop command in combination with a logical or conditional statement.” But you mentioned “Missing data were managed based on the DHS guideline” under the data source section. How do you relate those two sentences? Do you believe that “Removing” is considered as the management of missing data analysis? If not, manage your missing data based on imputation or other handling methods.

Authors’ Response:

Dear reviewer,

Thank you for your valuable comments. We have now included the version of STATA we used for the data analysis. We apologize for any confusion caused by our previous statement regarding missing data management. We adhered to the DHS guidelines, which led us to remove a small number of missing data points. These cases were minimal and did not impact the results, as allowed by the guidelines.

Reviewers’ Comments

Please thoroughly document all methods under the 'Spatial Scan Statistics' and 'Spatial Regression' sections, including relevant supporting references.

Authors’ Response:

Dear reviewer,

Thank you for your valuable feedback. We have thoroughly documented all methods under the 'Spatial Scan Statistics' and 'Spatial Regression' sections in the revised manuscript. This includes a comprehensive description of the spatial scan statistics and the spatial regression techniques applied in our study with relevant supporting references. We appreciate your time and consideration, and we trust the revisions meet your expectations.

Reviewers’ Comments

Please format the table title clearly (e.g., 'Problem, Population, Area, 2024, n=') and ensure the tables are cited throughout the document with cross-references.

Authors’ Response:

Dear reviewer,

Thank you for your comment. We have corrected the table title accordingly and the revised table title is as follows:

“Sociodemographic Characteristics of Mixed Milk Feeding Practices Among Infants Aged 0–5 Months in Ethiopia, EDHS 2019 (n = 524)”

Thank you again for your valuable feedback.

Reviewers’ Comments

Prevalence of mixed milk feeding in Ethiopia: “In this study, the overall prevalence of MMF was 10.12% (95% CI 7.8, 13.01), with regional variations observed across Ethiopia”. The results were not displayed in any tables or graphs. Please provide the prevalence data in either a table or a graph, and reference any tables used.

Authors’ Response:

Dear reviewer,

Thank you for your valuable suggestion. We have included a graph illustrating the prevalence of MMF across regions of Ethiopia. We believe this graph will enhance the manuscript.

Reviewers’ Comments

The 'Discussion' section can be generally divided into 3 separate paragraphs: 1) Introductory paragraph/rationale of the study, 2) Intermediate paragraphs/compare and contrast with the most recent and relevant literature, 3) Concluding paragraph/indicating future directions. Then address such things as well.

Authors’ Response:

Dear reviewer,

Thank you for your suggestions on structuring the discussion. We have divided it into an introduction, a comparison of findings, and a conclusion with future directions. We used multiple paragraphs for the comparison due to the need for detailed explanation. Thank you again for your valuable feedback.

Reviewers’ Comments

While the manuscript mentions the study utilized nationally representative data as a strength indicating there is adequate power to detect the true effect of exposures. However, it does not provide a power calculation to support this claim.

Authors’ Response:

Dear reviewer,

Thank you for your feedback. We did not conduct a formal power calculation for this study. Instead, we relied on the available sample, which is representative and uses complex survey design techniques to ensure robustness. Thank you Agin for your valuable feedback.

Reviewers’ Comments

Recommendation

This comprehensive report outlines the issues identified throughout the manuscript, with specific sections and requires substantial minor revisions to address these concerns and enhance the clarity, relevance, and presentation of results. After making the necessary minor revisions with re-write of manuscript, a re-evaluation is recommended for considering publication.

Authors’ response

Dear reviewer ,

Thank you for your reccomendation and the overall expertise feedback. We hope all the comments are addressed and we are very open to accept any additional comment and suggestions.

Thank you for your helpful comments, suggestions, and feedback for enhancing the manuscript. We are open to any further suggestions and look forward to incorporating them to improve our work.

---

## [Decision Letter · Decision Letter 1]

22 Dec 2024

Spatial distribution of mixed milk feeding and its determinants among mothers of infants aged under 6 months in Ethiopia: spatial and geographical weighted regression analysis.

PONE-D-24-28246R1

Dear Dr. Aweke,

We’re pleased to inform you that your manuscript has been judged scientifically suitable for publication and will be formally accepted for publication once it meets all outstanding technical requirements.

Kind regards,

Daniel Biftu Bekalo, PhD

Academic Editor

PLOS ONE

Additional Editor Comments (optional):

Reviewers' comments:

Reviewer's Responses to Questions

**Comments to the Author**

1. If the authors have adequately addressed your comments raised in a previous round of review and you feel that this manuscript is now acceptable for publication, you may indicate that here to bypass the “Comments to the Author” section, enter your conflict of interest statement in the “Confidential to Editor” section, and submit your "Accept" recommendation.

Reviewer #1: All comments have been addressed

2. Is the manuscript technically sound, and do the data support the conclusions?

Reviewer #1: Yes

3. Has the statistical analysis been performed appropriately and rigorously? 

Reviewer #1: Yes

4. Have the authors made all data underlying the findings in their manuscript fully available?

Reviewer #1: Yes

5. Is the manuscript presented in an intelligible fashion and written in standard English?

Reviewer #1: Yes

6. Review Comments to the Author

Reviewer #1: (No Response)

7. PLOS authors have the option to publish the peer review history of their article (what does this mean? ). If published, this will include your full peer review and any attached files.

**Do you want your identity to be public for this peer review?** For information about this choice, including consent withdrawal, please see our Privacy Policy .

Reviewer #1: No

---

## [Editor Report · Acceptance letter]

PONE-D-24-28246R1

PLOS ONE

Dear Dr. Aweke,

I'm pleased to inform you that your manuscript has been deemed suitable for publication in PLOS ONE. Congratulations! Your manuscript is now being handed over to our production team.

Kind regards,

on behalf of

Dr. Daniel Biftu Bekalo

Academic Editor

PLOS ONE